# Diffusion Tensor Imaging Radiomics for Diagnosis of Parkinson’s Disease

**DOI:** 10.3390/brainsci12070851

**Published:** 2022-06-29

**Authors:** Jingwen Li, Xiaoming Liu, Xinyi Wang, Hanshu Liu, Zhicheng Lin, Nian Xiong

**Affiliations:** 1Department of Neurology, Union Hospital, Tongji Medical College, Huazhong University of Science and Technology, Wuhan 430022, China; jingwenli1009@163.com (J.L.); wangxinyi_21@163.com (X.W.); liu1749676236@163.com (H.L.); 2Department of Radiology, Union Hospital, Tongji Medical College, Huazhong University of Science and Technology, Wuhan 430022, China; liuxiao880220@163.com; 3Hubei Province Key Laboratory of Molecular Imaging, Wuhan 430022, China; 4Laboratory of Psychiatric Neurogenomics, McLean Hospital, Harvard Medical School, Belmont, MA 02478, USA; zhicheng_lin@hms.harvard.edu; 5Wuhan Red Cross Hospital, Wuhan 430022, China

**Keywords:** DTI, Parkinson’s disease, radiomics, algorithm

## Abstract

Background: Diagnosis of Parkinson’s Disease (PD) based on clinical symptoms and scale scores is mostly objective, and the accuracy of neuroimaging for PD diagnosis remains controversial. This study aims to introduce a radiomic tool to improve the sensitivity and specificity of diagnosis based on Diffusion Tensor Imaging (DTI) metrics. Methods: In this machine learning-based retrospective study, we collected basic clinical information and DTI images from 54 healthy controls (HCs) and 56 PD patients. Among them, 60 subjects (30 PD patients and 30 HCs) were assigned to the training group, whereas the test cohort was 26 PD patients and 24 HCs. After the feature extraction and selection using newly developed image processing software Ray-plus, LASSO regression was used to finalize radiomic features. Results: A total of 4600 radiomic features were extracted, of which 12 were finally selected. The values of the AUC (area under the subject operating curve) in the training group, the validation group, and overall were 0.911, 0.931, and 0.919, respectively. Conclusion: This study introduced a novel radiometric and computer algorithm based on DTI images, which can help increase the sensitivity and specificity of PD screening.

## 1. Introduction

Parkinson’s disease (PD) is a common progressive neurological disorder with motor and cognitive disturbances, which can be characterized by the presence of Lewy bodies and a loss of dopaminergic neurons in the substantia nigra, resulting in disability and impaired life quality for patients [1]. Currently, the diagnosis of PD depends mainly on clinical examinations and partly on radiology [2]. It is challenging to make an accurate diagnosis of PD in the early stages.

In recent years, magnetic resonance imaging (MRI) technology has made significant progress in the field of neuroimaging. A variety of different functional imaging technologies represent effective methods for the non-invasive study of PD-related changes in brain morphology and function [3]. As a routine MRI imaging technique, diffusion tensor imaging (DTI) [4] has been widely used to provide quantitative diagnosis or prognosis in PD and other neurodegenerative diseases [5,6,7]. DTI is an in vivo diffusion imaging technique that reflects the diffusion of water molecules and microscopic changes in the white matter fiber bundles of the brain. Two indicators, fractional anisotropy (FA) and mean diffusivity (MD), are widely used to reveal the microstructure of normal and diseased tissues [8].

Although some studies have suggested that DTI appears to be a sensitive method for studying PD pathophysiology and severity [7], another study reported that DTI could not improve the diagnostic accuracy of brain MRIs for PD [9]. Therefore, in order to increase the accuracy of PD diagnosis, radiomics has emerged in recent years. Radiomics is a method for extracting high-dimensional quantitative features in images. It combines the spatial correlation of signal intensity in medical images and carries out subsequent data mining and application. Recent studies show that radiomics may detect imperceptible information reflecting common sequences, which is a potential and promising method for early identification and prognosis [10,11] and has been applied to the diagnosis of neurodegenerative diseases [12]. Previous radiomic studies made diagnoses and differential diagnoses mainly using routine T1- or T2-weighted MRIs [13,14]. In addition, it has been reported that DTI data could also be processed by a computer algorithm to evaluate brain connection and function [7].

In this study, we established a radiomics model based on DTI image data, aiming to identify some new related indicators to help clinical diagnosis. The findings show that radiomic technology, as a non-invasive tool, has the potential to support biopsies and be a supplement to previous radiodiagnoses. This study can thus help provide new clues for the early identification and treatment of PD patients with the achievement of the high sensitivity, specificity, and classification accuracy of the PD clinical diagnosis system.

## 2. Methods & Materials

### 2.1. Patients

The criterion for inclusion in the study was confirmed PD according to the Movement Disorder Society (MDS) clinical diagnostic criteria [15]. A total of 56 PD patients and 54 health controls (HCs) were enrolled in the study. Thirty patients and age/gender-matched HCs were enrolled in the training cohort. After the model was established, 24 HCs and 26 patients were included in the validation/test cohort. Characteristics of the two groups including age, gender, Unified Parkinson’s Disease Rating Scale (UPDRS)-Ⅲ, and Hoehn–Yahr (H&Y) stage of the training and validation cohorts are summarized in Table 1. The Institutional Review Board approved this retrospective study and we obtained informed consent from patients. A flow chart of the study is shown in Figure 1.

### 2.2. MRI Imaging Protocol

All PD patient and healthy control data were performed with a 3.0-T MRI scanner (Magnetom Trio System, Siemens Healthcare, Erlangen, Germany) with a 12-channel head coil. The head of each subject was positioned carefully with restraining pads to minimize head motions. High-resolution T1- and T2-weighted images were acquired for each participant to exclude the possibility of recessive lesions. DTI data were obtained using a single-shot echo-planar imaging sequence with the following parameters: TR = 6000 ms, TE = 93 ms, FA = 90°, 44 axial slices, slice thickness = 2mm, FOV = 256 × 256 mm^2^, matrix size = 128 × 128, noncollinear directions = 30, b-value = 1000 s/mm^2^ and b = 0 s/mm^2^ (no diffusion gradient), and scanning time = 6 min and 32 s. FA and MD maps were generated by a workstation (syngo.via XA20, Siemens Healthcare, Erlangen, Germany) after DTI acquisition and derived from the three eigenvalues (λ_1_, λ_2_, and λ_3_). The diffusion tensor was diagonalized to yield the major (λ_1_), intermediate (λ_2_), and minor (λ_3_) eigenvalues corresponding to the three eigenvectors in the diffusion tensor matrix [16]. MD was a voxel-wise measure of the directionally averaged magnitude of diffusion (unit: square millimeters per second), calculated as follows: MD = (λ_1_ + λ_2_ + λ_3_)/3 [17]. FA was used to measure the fraction of the total magnitude of diffusion, which was anisotropic, and had a value of 0 for isotropic diffusion (λ_1_ = λ_2_ = λ_3_) and 1 for completely anisotropic diffusion (λ_1_ ≥ 0, λ_2_ = λ_3_ = 0). FA was calculated as reported [18]:(1)$$\mathrm{FA}=\sqrt{\frac{3}{2}}\sqrt{\frac{{\left(\lambda_{1}-MD\right)}^{2}+{\left(\lambda_{2}-MD\right)}^{2}+{\left(\lambda_{3}-MD\right)}^{2}}{\lambda_{1}^{2}+\lambda_{2}^{2}+\lambda_{3}^{2}}},$$

### 2.3. MRI Image Segmentation and Feature Extraction

MRI image segmentation and radiomic feature extraction were performed with a newly developed image processing software Ray-plus (Rayplus Technology, Inc., Wuhan China) based on the software framework pyradiomics as described previously [17]. We firstly smoothed the images to fill the holes in the threshold segmentation process. The whole brain was set as the region of interest (ROI) using a threshold segmentation tool. For FA and MD images, positive signals mainly occurred inside the brain. Noise signals outside the brain were excluded manually. Feature extraction was performed using the ROI analysis module of Ray-plus (Figure 2). During the feature extraction process, we transformed the ROI to original, exponential, logarithm, wavelet, Gabor, and LoG forms. Radiomic features were extracted from six classes of matrices: histogram, volumetric, morphologic, the gray-level co-occurrence matrix (GLCM), the gray-level dependence matrix (GLDM), and the gray-level run-length matrix (GLRLM). For each sequence of ROI, we calculated 2300 features. Since each ROI had both FA and MD images, there were a total of 4600 radiomic features for each patient or HC.

### 2.4. Dimensionality Reduction

In order to avoid the curse of dimensionality caused by a large number of radiomic features, we performed dimensionality reduction in the training cohort in two steps. First, the radiomic features of all patients were each tested by a two-samples *t*-test or Mann–Whitney U test, which was recommended by the algorithm of the LASSO regression of the R project [19,20]. Features with significant differences (*p* < 0.001) between the PD group and the HC group were selected for further reduction. Second, the least absolute shrinkage and selection operator (LASSO) was used for regression and feature selection in the “glmnet” package of the R software (Version 3.4.1). An established statistical model enhanced the prediction accuracy and interpretability by performing variable selection and regularization. The min criteria (the min binomial deviance) were used to tune the regularization parameter (λ) in a 5-fold cross-validation for feature selection.

### 2.5. Radiomic Score Building

Logistic regression was used as the machine learning method to generate a combined radiomic score. Best predictable radiomic features were finally selected by dimensionality reduction. Multivariable binary logistic regression was used with the data of the training cohort and the coefficients of each feature were calculated. The assessment of the optimal radiomic score was performed using the receiver operating characteristic curve (ROC) system in the “pROC” package of the R software (Version 3.4.1). The area under the ROC (AUC) was used to evaluate the classification ability of the radiomic score (0.5–1.0, a higher value meant a better performance for diagnosis).

### 2.6. Statistics

Data were represented as mean ± SD. All statistical analyses were conducted using R software (Version 4.2.0, https://www.r-project.org/, accessed on 22 April 2022). LASSO regression based on multivariate binary logistic regression was performed with the “glmnet” package. ROC curves were created with the “pROC” package. Reported statistically significant differences were all two-sided, with statistical significance at *p* < 0.05.

## 3. Results

### 3.1. Patient Characteristics

We collected the basic information including the age and gender of the PD patients and HCs. In the training cohort, no significant difference in age was found between the PDs and the HCs (57.78 ± 7.68 and 57.69 ± 7.72, respectively, *p* = 0.2717), nor was there a significant difference in gender (15/15 and 14/16 in male/female, *p* > 0.9999). The clinical characteristics of the test group were also summarized. In addition, we have summarized the UPDRS and H&Y stages of the PD patients in the training and test cohorts in Table 1.

### 3.2. Feature Selection

Two (FA and MD) images of every patient were applied for radiomic features extraction. Because 2300 features were extracted from each image, there were 4600 features in total for one patient or HC. A total of 430 features were tested by two-sample *t*-tests or Mann–Whitney U tests and a significant difference was found between the PD and HC groups in the training cohort (*p* < 0.001). Therefore, the 430 features were used for the LASSO regression also in the training cohort. Seven features were selected by LASSO with the best-tuned regularization parameter λ found by fivefold cross-validation. The remaining seven features were used for the radiomic modeling (Figure 3).

### 3.3. Development of the Radiomic Score

In order to build an optimal predictive radiomic score, logistic regression was applied to generate a linear classifier. The radiomic score was developed from the seven radiomic features selected in the last step (Table 2). Based on the intercept and coefficients of each radiomic feature, the building formula of the radiomic score is shown in Figure 4.

### 3.4. Performance of the Radiomic Model

The AUC was generally used for the assessment of the classification and prediction ability of the radiomic models. Good performance based on the radiomic score was observed, as seen in Figure 5. The AUC of the training cohort was 0.911 and the AUC of the test cohort was 0.931. In total, the global data showed a high AUC of 0.919, indicating that the radiomics score generated by our radiomic analysis and modeling had a relatively high potential in predicting the occurrence of PD in the suspected patients.

## 4. Discussion

In this study, we have generated and evaluated a quantitative model based on the radiomic features extracted from the DTI images of patients’ brains for non-invasively predicting the occurrence of disease. In other words, we presented a convenient radiomics model based on DTI data (especially FA and MD features), with increased accuracy for an individual PD diagnosis. Compared to a previous study which reported an AUC of 0.733 in diagnosing PD [21], our predictive radiomic score showed excellent performance with a higher AUC in both the training and testing cohort (0.911 in the training cohort and 0.931 in the testing cohort), implying the clinical potential of radiomic scores in diagnosing PD patients in the early stages. A longitudinal correlation between the AUC and clinical symptoms in patients could be the desired data.

FA and MD are commonly used as DTI features and in clinical studies of PD patients [7,22]. A previous study found that there was no specific FA value with high sensitivity and specificity for both the screening and diagnosis of PD [6]. In addition, a recent study of DTI introduced the architecture of a convolutional neural network to distinguish PD patients from HCs; however, the algorithm was much more complex [23]. Therefore, in our study, we proposed a convenient model based on FA and MD values to improve the accuracy of PD diagnoses. One of the main advantages of our radiomics model is that we set the whole brain as the ROI to enhance its applicability in wide clinical use. Different regions of the brain were previously chosen as the ROI for feature extraction [24]. However, the segmentation of the given regions in the brain faces difficult challenges in precise delineation. Usually, at least two experienced radiologists are required to conform to the same ROI of each patient, which greatly limits the application of the radiomics models. Normalizing the patient brain images to a standard brain templet is another method for accurate dilation of brain regions, but vast information related to shape was lost due to deformations in the normalization. Setting the whole brain as the ROI ensured the convenience and reliability of our radiomic model. After the calculation of the algorithm, seven features of FA maps were finally screened for the model establishment, which may be because of the same sources of the FA and MD maps with similar commonalities and connections generated by the DTI images. Then we can better distinguish PD patients individually through a small amount of data while making important clinical predictions.

There are a few limitations to our study. First, this was a retrospective study from a single site with a limited sample size. To solve this limitation, we used cross-validation and independent test sets for internal validation. Specifically, N-fold cross-validation was used in the LASSO regression with subsequent validation groups as independent internal validation groups [25]. Second, the small sample size and single DTI image omics lead to the limited diagnostic efficacy in differentiating different PD types. In addition, the b-matrix spatial distribution in the DTI (BSD-DTI) method could be used in future investigations [26,27,28,29].

Overall, our study proposes a radiomics model based on DTI that can improve the specificity and sensitivity of the diagnosis of neurodegenerative diseases including PD. The identification of such new biomarkers is crucial for promoting the early recognition and a better understanding of the pathogenesis of PD.

## Figures and Tables

**Figure 1 brainsci-12-00851-f001:**
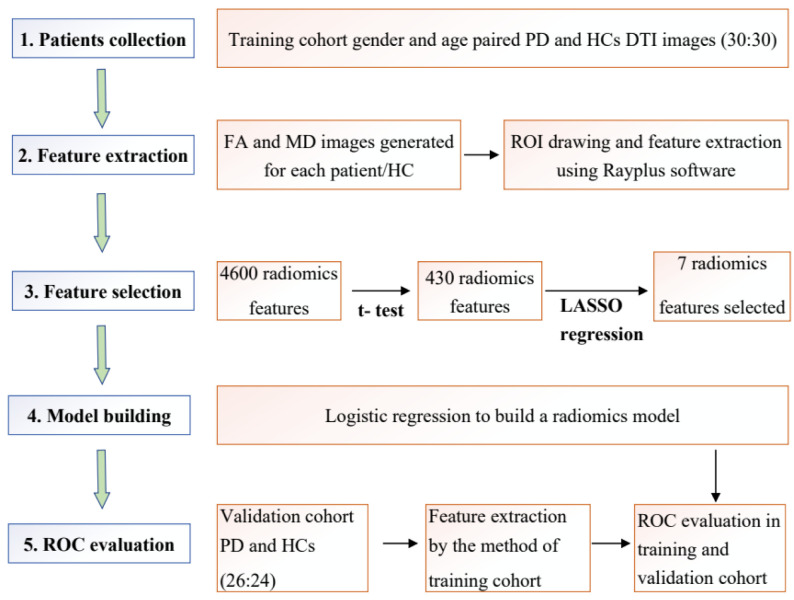
Flow chart for the whole study.

**Figure 2 brainsci-12-00851-f002:**
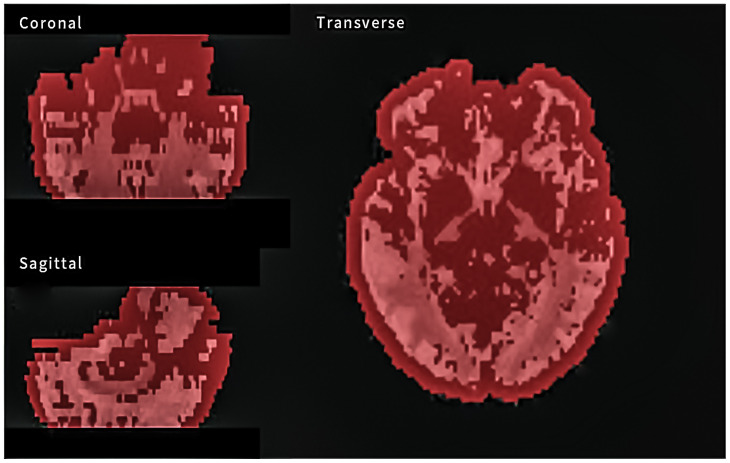
MRI image segmentation feature extraction by Ray-plus software (the whole brain was selected as the ROI for feature extraction in red; grey means FA/MD values).

**Figure 3 brainsci-12-00851-f003:**
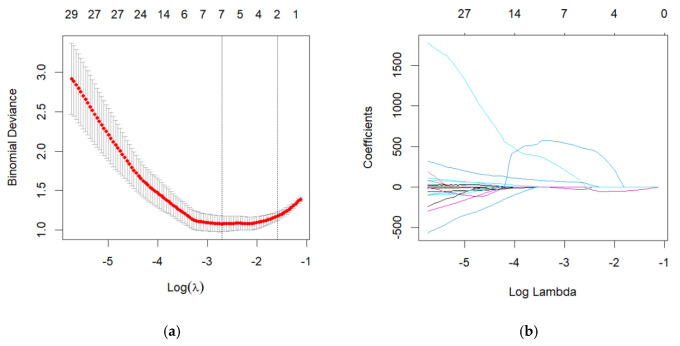
Radiomic feature selection using the least absolute shrinkage and selection operator (LASSO) binary logistic regression model. (**a**) Tuning parameter (λ) selection in the LASSO model used 5-fold cross-validation via minimum criteria. The binomial deviance was plotted versus log (λ). Dotted vertical lines were drawn at the optimal values using the minimum criteria and the λ standard error of the minimum criteria (the 1−SE criteria). According to 5-fold cross-validation, the minimum criteria (λ = 0.06656, log (λ) = −2.700) were chosen for selection of radiomic features. (**b**) LASSO coefficient profiles of the 7 selected features. A coefficient profile plot was produced against the log (λ) sequence.

**Figure 4 brainsci-12-00851-f004:**
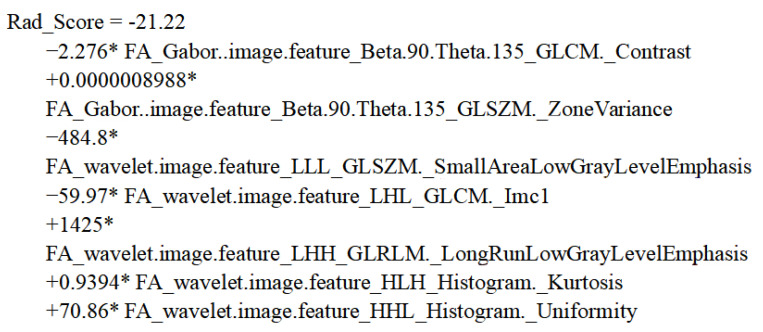
Formula of radiomic score.

**Figure 5 brainsci-12-00851-f005:**
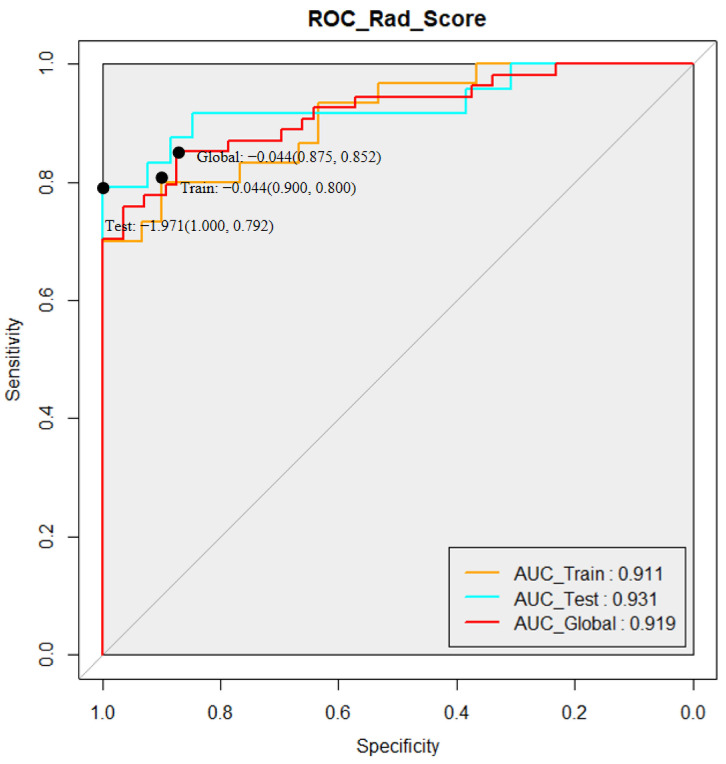
Sensitivity and specificity of the radiomic model. Data was represented as Rad_score (specificity, sensitivity).

**Table 1 brainsci-12-00851-t001:** Basic characteristics of patients and controls.

	Train Cohort			Test Cohort			HC-Train vs. PD-Train *p* Value
	PD (30)	HC (30)	All (60)	PD (26)	HC (24)	All (50)	
Age (years, mean ± SD)	57.78 ± 7.68	57.69 ± 7.72	57.83 ± 7.63	65.12 ± 12.54	38.71 ± 11.26	52.44 ± 17.79	0.2717
Gender (male/female)	15/15	14/16	29/31	17/9	12/12	29/21	>0.999
UPDRS	29.87 ± 14.62	/	/	32.75 ± 17.36	/	/	/
H&Y stage	1.83 ± 0.77	/	/	2 ± 0.71	/	/	/

**Table 2 brainsci-12-00851-t002:** List of maps, source (image form), algorithm of source, feature class, feature name, and formula of each selected radiomic feature.

Feature	Maps	Source	Algorithm	Class	Feature	Equation
No.172	FA	Gabor	Beta.90.Theta.135	GLCM	Contrast	$\overset{N_{g}}{\underset{i=1}{\sum }}\overset{N_{g}}{\underset{j=1}{\sum }}{\left(i-j\right)}^{2}p\left(i,j\right)$
No.191	FA	Gabor	Beta.90.Theta.135	GLSZM	ZoneVariance	$\overset{N_{g}}{\underset{i=1}{\sum }}\overset{N_{g}}{\underset{j=1}{\sum }}p\left(i,j\right){\left(j-\mu \right)}^{2}$
No.242	FA	wavelet	LLL	GLSZM	SmallAreaLowGrayLevelEmphasis	$\frac{\sum_{i=1}^{N_{g}}\sum_{j=1}^{N_{s}}\frac{P_{\left(i,j\right)}}{i^{2}j^{2}}}{N_{z}}$
No.297	FA	wavelet	LHL	GLCM	lmc1	$\frac{HXY-HXY1}{max\left\{HX,HY\right\}}$
No.320	FA	wavelet	LHH	GLRLM	LongRunLowGrayLevelEmphasis	$\frac{\sum_{i=1}^{N_{g}}\sum_{j=1}^{N_{r}}\frac{P(i,j|\theta )j^{2}}{i^{2}}}{N_{r}\left(\theta \right)}$
No.360	FA	wavelet	HLH	Histogram	Kurtosis	$\frac{\frac{1}{N_{p}}\sum_{i=1}^{N_{p}}{\left(X\left(i\right)-\bar{X}\right)}^{4}}{{\left(\frac{1}{N_{p}}\sum_{i=1}^{N_{p}}{\left(X\left(i\right)-\bar{X}\right)}^{2}\right)}^{2}}$
No.391	FA	wavelet	HHL	Histogram	Uniformity	$\overset{N_{g}}{\underset{I=1}{\sum }}p{\left(i\right)}^{2}$

## Data Availability

All relevant data generated during this study are available from the corresponding author on reasonable request.

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
