# Peer review of "Diffusion Tensor Imaging Radiomics for Diagnosis of Parkinson’s Disease"

_brainsci, 2022, doi:10.3390/brainsci12070851_

Round 1

Reviewer 1 Report

Summary:
The authors tried to classify PD and HC groups using an existing radiomics tool. The study do not look significant and not interesting due to the lack of any novel contribution or findings.

Review Comments:
Introduction:
1. The reason or motivation behind the study to classify PD group is not discussed. While the authors say T1 and T2 are mostly used for diagnosing PD with poor sensitivity and specificity without proper literature, I do see previous works reported on PD diagnosis using diffusion MRI (Zhao 2022). The authors failed to discuss the real state of the art and  the gaps and the motivation behind trying radiomics for this hypothesis.

Zhao2022: Zhao H, Tsai CC, Zhou M, Liu Y, Chen YL, Huang F, Lin YC, Wang JJ. Deep learning based diagnosis of Parkinson's Disease using diffusion magnetic resonance imaging. Brain Imaging Behav. 2022 Mar 14. doi: 10.1007/s11682-022-00631-y. Epub ahead of print. PMID: 35285004.

Methods:
2. The methods section looks incompletely written with technical details missing for all the processing steps in the classification study
3. The technical details about the radiomics toolbox related to feature generation is lacking. What radiomics features are used, the authors have to broadly list the category of features included for the study
4. How is the whole brain segmentation done, what algorithm is used?
5. Figure 1 does not convey any information, this looks like a screenshot from the toolbox with non-English titles. What do the color intensity denote?
6. DTI features such as Mean/Axial/Radial diffusivity were shown to have better diagnostic accuracy as compared to conventional ADC, the authors need to discuss the reason behind selecting ADC.
7. Dimensionality reduction: what is the reason behind selecting t-test for feature reduction despite advanced methods for dimensionality reduction?
8. How is 3 fold cross validation selected for this study?

Results:
Line 133 - 140: The differences of clinical variable between the 2 groups are not discussed. This would give idea on the influential factors apart from MRI features to include as covariates in statistical analysis
Figure 2: Details about figure 2 are not clear, legend missing

Discussion:
Why do radiomics result in better accuracy, how is it related to PD?

Author Response

Summary:
The authors tried to classify PD and HC groups using an existing radiomics tool. The study do not look significant and not interesting due to the lack of any novel contribution or findings.

Thank you. MRI including DTI has been widely used as a common radiomic tool in many diseases such as PD. The accuracy of neuro-imaging for PD diagnosis remains controversial. Some studies suggested DTI appears to be a sensitive method to study PD pathophysiology and severity while another study reported that DTI could not improve the diagnostic accuracy of brain MRI to identify PD[1]. In this study, we introduced a novel radiometrics and computer algorithm based on DTI, which could increase the higher accuracy of PD diagnosis (AUC: 0.919, comparing to a previous model with AUC 0.733). We have clarified this in the discussion part.

Review Comments:
Introduction:
1. The reason or motivation behind the study to classify PD group is not discussed. While the authors say T1 and T2 are mostly used for diagnosing PD with poor sensitivity and specificity without proper literature, I do see previous works reported on PD diagnosis using diffusion MRI (Zhao 2022). The authors failed to discuss the real state of the art and the gaps and the motivation behind trying radiomics for this hypothesis.

Zhao2022: Zhao H, Tsai CC, Zhou M, Liu Y, Chen YL, Huang F, Lin YC, Wang JJ. Deep learning based diagnosis of Parkinson's Disease using diffusion magnetic resonance imaging. Brain Imaging Behav. 2022 Mar 14. doi: 10.1007/s11682-022-00631-y. Epub ahead of print. PMID: 35285004.

Thanks for your review. We’ve revised the introduction into “As a routine MRI imaging technique, diffusion tensor imaging (DTI) has been widely used to provide quantitative diagnosis or prognosis in PD and other neurodegenerative diseases[2].  However, previous study reported that DTI did not improve the diagnostic accuracy of brain MRI for PD1. Therefore, in order to increase the accuracy, a relatively new and powerful technical means, – radiomics, has emerged [3, 4].” on page 3. This study was to apply such new means in diagnosis with PD.

Methods
2. The methods section looks incompletely written with technical details missing for all the processing steps in the classification study.

We have re-written the methods part into “In this toolbox, we performed the feature generation method using the software Rayplus1 based on the software framework pyradiomics as described previously[5]. We firstly smoothed the images to fill the holes in the threshold segmentation process. And the whole brain was set as the region of interest (ROI) using a threshold segmentation tool. During the feature extraction process, we transformed the ROI to original, exponential, logarithm, wavelet, Gabor and LoG forms. Radiomics features were extracted from six classes of matrices: histogram, volumetric, morphologic, the gray-level co-occurrence matrix (GLCM), the gray-level dependence matrix (GLDM) and the gray-level run length matrix (GLRLM). For each sequence of ROI, we calculated 2300 features. As each ROI had both FA and MD images, there were 4600 radiomics features for each patient or HC. ” See page 5 in text.

  1. The technical details about the radiomics toolbox related to feature generation is lacking. What radiomics features are used, the authors have to broadly list the category of features included for the study.

      Please see above. We have re-written the methods part and completed the          technical details in the revised manuscript on page 5.

  1. How is the whole brain segmentation done, what algorithm is used?

    We used threshold segmentation method to set the whole brain as ROI.          Before that, we smoothed the images to fill the holes in the threshold segmentation process. We also added this sentence into the revised manuscript now (page 5, lines 132-134).

  1. Figure 1 does not convey any information, this looks like a screenshot from the toolbox with non-English titles. What do the color intensity denote?

The gray color intensity denoted FA and MD values. The red color indicated the area of ROI. We’ve also added this in the title of Figure 1 (now presented as Figure 2).

  1. DTI features such as Mean/Axial/Radial diffusivity were shown to have better diagnostic accuracy as compared to conventional ADC, the authors need to discuss the reason behind selecting ADC.

Thank you for your suggestions. In order to have better diagnostic accuracy, we replaced ADC with MD and carried out re-analysis in the revised manuscript.

  1. Dimensionality reduction: what is the reason behind selecting t-test for feature reduction despite advanced methods for dimensionality reduction?

According to previous publications[6, 7], we selected t-test to exclude the features with no significance and reduce the feature number (under 500), which was recommended by the algorithm of LASSO regression of R project. This is clarified page 6.

  1. How is 3 fold cross validation selected for this study?

Previously, we selected 3-fold cross because of the small total number of PD patients. We were worried that the model effect would be affected by the shortage of patients in a single cross-grouping in the case of 5-fold validation. And in the revised manuscript, we have re-designed our workflow in analysis and adapted the widely used 5-fold cross validation in the LASSO regression.

Results:
Line 133 - 140: The differences of clinical variable between the 2 groups are not discussed. This would give idea on the influential factors apart from MRI features to include as covariates in statistical analysis.

Thanks for your suggestions. Now we added the clinical information (UPDRS and Hoehn-yahr(H&Y) stage) of PD patients in Table 1 (page 7).

Figure 2: Details about figure 2 are not clear, legend missing.

We’ve added the figure legend (now presented as Figure 3) as “Left: Tuning parameter (λ) selection in the LASSO model used 5-fold cross-validation via minimum criteria. The binomial deviance was plotted versus log(λ). Dotted vertical lines were drawn at the optimal values by using the minimum criteria and the λ standard error of the minimum criteria (the 1-SE criteria). According to 5-fold cross-validation, the minimum criteria(λ = 0.06656, log (λ) = -2.700) was chosen for selection of radiomisc features. Right: LASSO coefficient profiles of the 7 selected features. A coefficient profile plot was produced against the log (λ) sequence.”.

Discussion:
Why do radiomics result in better accuracy, how is it related to PD?

Currently, the diagnosis of PD basically relies on clinical symptoms and scale evaluation, which are mostly subjective. DTI features including FA, MD and some other values are commonly used in clinical studies of PD patients[8-10]. In this study, we generated and evaluated a quantitative radiomics score based on the radiomics features extracted from the DTI images of patients’ brain for non-invasively predicting the occurrence of disease. In other words, we presented a novel radiometrics and computer algorithm based on DTI (especially FA and MD sequences), with an increased accuracy of PD diagnosis. Comparing to the reported AUC of 0.733 in diagnosing PD[11], our predictive radiomics score showed excellent performance with a higher AUC in both the training and testing cohort (0.911 in the training cohort and 0.931 in the testing cohort), implying the clinical potential of radiomics score in diagnosing the PD patients in an early stage. ” We’ve also added this in the discussion part (page 9).

Reviewer 2 Report

This paper introduced a machine learning method to diagnose Parkinson’s disease using features extracted from Diffusion Tensor Imaging (DTI), so called radiomics, The paper shows that the trained model is capable of distinguish patients from healthy control subjects. As mentioned by the authors, so far there is no reliable tool for early diagnosis of PD, and the results presented in this paper seems encouraging. However, some details about the methodology are missing, and some grammar errors need to be fixed. Below are my specific comments.

Major comments:

  1. Since the aim of this study is to show the capability of DTI radiomcis for early diagnosis of PD, the authors should specify the severity or stages/levels of the PD patients included in this paper.
  2. A typical cross-validation fold number is 5 or 10. Why was a 3-fold CV used?
  3. The age distributions of PD and HC seems to be very different. This may cause statistical bias and lead to unreliable machine learning model.

Minor comments:

  1. Last sentence of the Abstract: There should be a “which” between “DTI images” and “can”.
  2. Line 38, Page 1. “It is challenging to make accurate diagnosis in the early stages of PD to avoid late complications.” This is unclear. Complications of what?
  3. Line 43, Page 1. T1 and T2 weighted MRI are structural image methods, not functional.
  4. Lin 87, Page 2. What does “bindicated” mean here?
  5. Line 90, Page 2. λ1= >0 should be λ1>=0, or simply λ1>0 ?
  6. The original publications about DTI should be cited. For example, Basser PJ, Mattiello J, LeBihan D. MR diffusion tensor spectroscopy and imaging. Biophys J 1994;66:259–267.
  7. More references about PD diagnosis using MRI should be provided.
  8. Table 2. The meaning of the equations, as well as the symbols (Np, Nz, etc) should be introduced.

Author Response

This paper introduced a machine learning method to diagnose Parkinson’s disease using features extracted from Diffusion Tensor Imaging (DTI), so called radiomics, The paper shows that the trained model is capable of distinguish patients from healthy control subjects. As mentioned by the authors, so far there is no reliable tool for early diagnosis of PD, and the results presented in this paper seems encouraging. However, some details about the methodology are missing, and some grammar errors need to be fixed. Below are my specific comments.

Major comments:

  1. Since the aim of this study is to show the capability of DTI radiomcis for early diagnosis of PD, the authors should specify the severity or stages/levels of the PD patients included in this paper.

Thanks for your suggestions. We’ve added the severity or stages/levels of the PD patients (UPDRS and Hoehn-yahr stages) in Table 1.

  1. A typical cross-validation fold number is 5 or 10. Why was a 3-fold CV used?

Thanks for your suggestions. 5-fold cross validation is used now instead.

  1. The age distributions of PD and HC seems to be very different. This may cause statistical bias and lead to unreliable machine learning model.

    Thank you for pointing out this. To avoid the big age distributions of PD and HC, we re-divided the cohort in order to make the age/gender matched and did a statistic analysis again in the training cohort (TABLE 1) and the results showed no significant difference in age distribution(P>0.05).

Minor comments

  1. Last sentence of the Abstract: There should be a “which” between “DTI images” and “can”.

   Thanks. we’ve added “which” between “DTI images” and “can”.

  1. Line 38, Page 1. “It is challenging to make accurate diagnosis in the early stages of PD to avoid late complications.” This is unclear. Complications of what?

   Thank you. We’ve revised the sentence into “It is challenging to make accurate diagnosis of PD in the early stages”.

  1. Line 43, Page 1. T1 and T2 weighted MRI are structural image methods, not functional.

   Thank you. We’ve deleted the sentence in the revised manuscript.

  1. Lin 87, Page 2. What does “bindicated” mean here?

   Thanks for your careful review and we’ve revised it into “indicated”.

  1. Line 90, Page 2. λ1= >0 should be λ1>=0, or simply λ1>0 ?  

Thanks for your careful review and we’ve revised it into “λ1≥0”.

  1. The original publications about DTI should be cited. For example, Basser PJ, Mattiello J, LeBihan D. MR diffusion tensor spectroscopy and imaging. Biophys J 1994;66:259–267.

Thank you and we’ve added this reference (REF 4) and other related ones in the revised manuscript.

  1. More references about PD diagnosis using MRI should be provided.

   Thank you and we’ve added more reference about PD diagnosis using MRI in the revised manuscript, including refs 5, 9, 10, 11, 12 and 22.

  1. Table 2. The meaning of the equations, as well as the symbols (Np, Nz, etc) should be introduced.

We’ve revised the title of Table 2 into “Table 2. The list of maps, source (image form), Algorithm of source, feature class, feature name and formula of each selected radiomics feature”.

Reviewer 3 Report

Review

The authors of the publication: Diffusion Tensor Imaging Radiomics for Diagnosis of Parkinson's Disease, present an interesting paper that shows the potential of using the features of radiomics obtained from DTI for the diagnosis of Parkinson's disease.

The vision is interesting and seems perspective, also for other dementia diseases. The article is written quite unevenly and contains significant deficiencies that seem to be possible to fill in at the manuscript revision stage. I will mention the most important issues.

1. The description of the parameters obtainable from the DTI should be described in more detail,
and the authors should justify the selection of the FA and ADC from the group of possible /multiple parameters. It is especially not entirely obvious to select an ADC; according to the givenformula, is it just the projection of the diffusion tensor onto a single direction of the diffusion gradient vector? What about the trace-Tr of the diffusion tensor, or MD, ....

2. The formation and selection of radiomics features is not clearly described. Starting from the idea itself and how to implement it in this case. It is described in the article very generally, far too general, giving the impression that the authors did not fully understand what they were writing about. 

There are terms that are not well explained, ROC curve, pROC packet, LASSO algorithm, ....

3. The method of data preparation and description remains a separate issue. Unfortunately, there is no information about the possible impact on the obtained results of systematic errors, which in the case of the DTI technique are possible and may be significant. I suggest discussing this issue in the context of the BSD-DTI method, in terms of detection and evaluation of possible systematic errors on Diffusion Tensor Imaging parameters. Quite abundant information on this issue can be found in the literature.

These are the most important issues, in my opinion, which, after supplementing, explaining and conducting a discussion, based on relevant items from the literature, should lead to the publication of the manuscript.

The article is also written linguistically unevenly, some parts are described correctly, others very vague, especially the chapters on radiomics. Overall, it might be an interesting article and I see the possibility of publishing this manuscript, however in my opinion it requires more care and I hope that the comments provided will help to achieve this diligence.

Author Response

The authors of the publication: Diffusion Tensor Imaging Radiomics for Diagnosis of Parkinson's Disease, present an interesting paper that shows the potential of using the features of radiomics obtained from DTI for the diagnosis of Parkinson's disease.

The vision is interesting and seems perspective, also for other dementia diseases. The article is written quite unevenly and contains significant deficiencies that seem to be possible to fill in at the manuscript revision stage. I will mention the most important issues.

  1. The description of the parameters obtainable from the DTI should be described in more detail, and the authors should justify the selection of the FA and ADC from the group of possible /multiple parameters. It is especially not entirely obvious to select an ADC; according to the given formula, is it just the projection of the diffusion tensor onto a single direction of the diffusion gradient vector? What about the trace-Tr of the diffusion tensor, or MD, ....

Thank you for your suggestions. In the revised manuscript, we replaced ADC with MD and made re-analysis using corresponding formula in the revised manuscript. And we added the details about feature extraction in the methods.

  1. The formation and selection of radiomics features is not clearly described. Starting from the idea itself and how to implement it in this case. It is described in the article very generally, far too general, giving the impression that the authors did not fully understand what they were writing about. 

There are terms that are not well explained, ROC curve, pROC packet, LASSO algorithm, ....

Thanks for your review. We’ve revised the methods part in current manuscript “We firstly smoothed the images to fill the holes in the threshold segmentation process. And the whole brain was set as the region of interest (ROI) using a threshold segmentation tool. For FA and MD images, positive signals mainly occurred inside the brain. Noise signals outside the brain were excluded manually. Feature extraction was performed using the ROI analysis module of Ray-plus (Figure 2). During the feature extraction process, we transformed the ROI to original, exponential, logarithm, wavelet, Gabor and LoG forms. The radiomics features were extracted from six classes of matrices: histogram, volumetric, morphologic, the gray-level co-occurrence matrix (GLCM), the gray-level dependence matrix (GLDM) and the gray-level run length matrix (GLRLM). For each sequence of ROI, we calculated 2300 features. As each ROI had both FA and MD images, there were 4600 radiomics features for each patient or HC. ” on page 5.

  1. The method of data preparation and description remains a separate issue. Unfortunately, there is no information about the possible impact on the obtained results of systematic errors, which in the case of the DTI technique are possible and may be significant. I suggest discussing this issue in the context of the BSD-DTImethod, in terms of detection and evaluation of possible systematic errors on Diffusion Tensor Imaging parameters. Quite abundant information on this issue can be found in the literature.

Thanks for your advice. We’ve added this in the discussion part. “ In order to evaluate and avoid possible systematic errors on DTI parameters, b-matrix Spatial Distribution in DTI (BSD-DTI) method could be used in future investigations[1-2]. ” (page 9).

These are the most important issues, in my opinion, which, after supplementing, explaining and conducting a discussion, based on relevant items from the literature, should lead to the publication of the manuscript.

The article is also written linguistically unevenly, some parts are described correctly, others very vague, especially the chapters on radiomics. Overall, it might be an interesting article and I see the possibility of publishing this manuscript, however in my opinion it requires more care and I hope that the comments provided will help to achieve this diligence.

Thank you. We’ve revised the full-text with revisions highlighted in blue color.

References:

[1]. Borkowski, K. and A.T. Krzyzak, Analysis and correction of errors in DTI-based tractography due to diffusion gradient inhomogeneity. J Magn Reson, 2018. 296: p. 5-11.

[2]. Borkowski, K. and A.T. Krzyzak, Assessment of the systematic errors caused by diffusion gradient inhomogeneity in DTI-computer simulations. NMR Biomed, 2019. 32(11): p. e4130.

Round 2

Reviewer 1 Report

The authors response for most of the questions do not completely address the concerns raised in the question. The authors responses partially answered or did not address the actual question, specifically for questions 1, 5,6,7 and the discussion section related question.

Introduction: When the authors mention "In this study, we present a novel computer algorithm...", I would suggest not to use words like "novel algorithm", which may give the readers a wrong understanding, whereas in the work, the authors actually tried to apply existing methods available in open source tools on diffusion data to experiment the accuracy in PD diagnosis.

On the whole, I would suggest, the authors should focus more on writing the introduction and discussion part to explain what has been really tried in this work and to justify the motivation behind the work by doing an extensive literature survey to understand the state of the art, without this the work does not show any significance in its present state.

Author Response

The authors response for most of the questions do not completely address the concerns raised in the question. The authors responses partially answered or did not address the actual question, specifically for questions 1, 5, 6, 7 and the discussion section related question.

  1. The reason or motivation behind the study to classify PD group is not discussed. While the authors say T1 and T2 are mostly used for diagnosing PD with poor sensitivity and specificity without proper literature, I do see previous works reported on PD diagnosis using diffusion MRI (Zhao 2022). The authors failed to discuss the real state of the art and the gaps and the motivation behind trying radiomics for this hypothesis.

Zhao2022: Zhao H, Tsai CC, Zhou M, Liu Y, Chen YL, Huang F, Lin YC, Wang JJ. Deep learning based diagnosis of Parkinson's Disease using diffusion magnetic resonance imaging. Brain Imaging Behav. 2022 Mar 14. doi: 10.1007/s11682-022-00631-y. Epub ahead of print. PMID: 35285004.

Thanks for your review. We’ve clarified the introduction. “Previous radiomics studies made the diagnosis and differential diagnosis mainly by routine T1 or T2-weighted MRI. In addition, it has been reported that DTI data could also be processed by computer algorithm to evaluate brain connection and function. In this study, we estabilsed a radiomics model based on DTI image data, aiming to identify some new related indicators to help clinical diagnosis.” on page 3.

  1. Figure 1 does not convey any information, this looks like a screenshot from the toolbox with non-English titles. What do the color intensity denote?

We’re revised the description of Figure 1 (now presented as Figure 2). The red color indicated the area of ROI. The grey color intensity denoted FA and MD values.

  1. DTI features such as Mean/Axial/Radial diffusivity were shown to have better diagnostic accuracy as compared to conventional ADC, the authors need to discuss the reason behind selecting ADC.

Thank you for your suggestions. In the current version of manuscript, we used MD instead of ADC, and the results showed better diagnostic accuracy (AUC).

  1. Dimensionality reduction: what is the reason behind selecting t-test for feature reduction despite advanced methods for dimensionality reduction?

We selected t-test with reference to previous publications, to exclude the features with no significance and reduce the feature number (under 500), which was recommended by the algorithm of LASSO regression of R project.

Discussion:
Why do radiomics result in better accuracy, how is it related to PD?

In this study, we have generated and evaluated a quantitative model based on the radiomic features extracted from the DTI images of patients’ brain for non-invasively predicting the occurrence of disease. In other words, we presented a convenient radiomics model based on DTI data (especially FA and MD features), with an increased accuracy of PD diagnosis. Comparing to a previous study which reported the AUC of 0.733 in diagnosing PD, our predictive radiomic score showed excellent performance with a higher AUC in both the training and testing cohort (0.911 in the training cohort and 0.931 in the testing cohort), implying the clinical potential of radiomic score in diagnosing the PD patients in an early stage.

FA and MD are commonly used as DTI features and used in clinical study of PD patients. A previous study found that there was no specific FA value with a high sensitivity and specificity for both screening and diagnosis of PD. In addition, a recent study of DTI introduced an architecture of convolutional neural network to distinguish PD patients from HCs, however, the algorithm was much complex. Therefore, in our study, we proposed a convenient model based on FA and MD values to improve the accuracy of PD diagnosis. One of the main advantages of our radiomics model is that we set the whole brain as the ROI to enhance the applicability in wide clinical use.

We’ve also added this in the discussion part.

Introduction: When the authors mention "In this study, we present a novel computer algorithm...", I would suggest not to use words like "novel algorithm", which may give the readers a wrong understanding, whereas in the work, the authors actually tried to apply existing methods available in open source tools on diffusion data to experiment the accuracy in PD diagnosis.

Thanks for your review. We’ve changed that description into “we presented a convenient radiomics model based on DTI data (especially FA and MD features), with an increased accuracy of PD diagnosis.” in both introduction and discussion.

On the whole, I would suggest, the authors should focus more on writing the introduction and discussion part to explain what has been really tried in this work and to justify the motivation behind the work by doing an extensive literature survey to understand the state of the art, without this the work does not show any significance in its present state.

Thank you. We’ve re-organized the introduction and discussion parts in the revised manuscript (with revisions highlighted in red color).

Reviewer 2 Report

I would like to thank the author for addressing my comments. I have no further critiques.

Author Response

Thank you for the review.

Reviewer 3 Report

Dear Authors,

Article seems publishable.

I also suggest the final "tuning" in terms of:

- a small discussion of the usefulness of individual DTI parameters for the construction of the radiomic model.

- additional, short literary extension of the BSD-DTI context with items showing the theoretical foundations (the generalized Stejskal-Tanner equation for non-uniform magnetic gradients; theoretical validation of BSD-DTI ..).

Additionally, I suggest linguistic correction in the context of sentence structure (clarity).

Good luck with your further research.

Author Response

Dear Authors,

Article seems publishable.

I also suggest the final "tuning" in terms of:

- a small discussion of the usefulness of individual DTI parameters for the construction of the radiomic model.

Thank you. We’ve added the discussion of DTI parameters “ FA and MD are commonly used as DTI features and used in clinical study of PD patients. A previous study found that there was no specific FA value with a high sensitivity and specificity for both screening and diagnosis of PD. In addition, a recent study of DTI introduced an architecture of convolutional neural network to distinguish PD patients from HCs, however, the algorithm was much complex. Therefore, in our study, we proposed a convenient model based on FA and MD values to improve the accuracy of PD diagnosis. One of the main advantages of our radiomics model is that we set the whole brain as the ROI to enhance the applicability in wide clinical use. In previous reports, different regions of brain was chosen as the ROI for feature extraction. However, the segmentation of the given regions in brain faces difficult challenges in precisely delineation. Usually, at least two experienced radiologists were required to conform the same ROI of each patient, which greatly limited the application of radiomics models. Normalize the patient brain images to standard brain templet is another method for accurate dilation of brain regions, but vast information related to shape was lost due to the deformation in normalization. Setting the whole brain as the ROI ensured the convenience and reliability of our radiomic model. After the calculation of algorithm, 7 features of FA maps were finally screened out for model establishment, which may because of the same sources of FA and MD maps, with similar commonalities and connections generated by DTI images. Then we can better distinguish PD patients individually through small amount of data while making important clinical predictions.” on page 9.

- additional, short literary extension of the BSD-DTI context with items showing the theoretical foundations (the generalized Stejskal-Tanner equation for non-uniform magnetic gradients; theoretical validation of BSD-DTI ..).

Thank you. We added more references about BSD-DTI in the revised version to make a better understanding (ref 30-33).

Additionally, I suggest linguistic correction in the context of sentence structure (clarity).

Thank you. We have sent the manuscript to a native speaker for further editing.
